# A Preference-Driven Smart Home Service for the Elderly’s Biophilic Experience

**DOI:** 10.3390/s21155108

**Published:** 2021-07-28

**Authors:** Eun-Ji Lee, Sung-Jun Park

**Affiliations:** 1Department of Architecture, Keimyung University, Daegu 42601, Korea; yej@stu.kmu.ac.kr; 2Department of Architectural Engineering, Keimyung University, Daegu 42601, Korea

**Keywords:** smart home service, elderly population, mental health, biophilic experience, smart technology

## Abstract

Smart home services (SHS) should support the positive experiences of the elderly in homes with a focus on getting closer to nature. The study identified the services preferred by the elderly through a survey on the biophilic experience-based SHS, and to discuss the configuration of the sensors and devices required to provide the service. We reorganized the biophilic experience-based SHS and related sensors and devices, focusing on our previous study, and developed a survey instrument. A preference survey was conducted on 250 adults aged 20 and older, and the SPSS program was used for a factor analysis and independent two-sample T-test. We derived six factors for biophilic experience-based SHS. Compared to other age groups, the elderly preferred services that were mainly attributed to factors such as ‘Immersion and interaction with nature’ (A), ‘Management of well-being and indoor environmental quality (IEQ)’ (B), and ‘Natural process and systems’ (F). We proposed 15 prioritized services, along with their sensor and device configurations, in consideration of service provision regarding the elderly’s preferences and universality. This study contributes to new developments in elderly-friendly smart home research by converting bio-friendly ideas into the market in the development of medical services and SHS for the elderly.

## 1. Introduction

The global aging population has placed new demands for families, communities, and governments, including health management, leisure culture, and a sustainable well-being environment. In particular, the connection between the level of physical, mental, social, and economic support in the architectural space and quality of life (QoL) in old age is continuously emphasized. The socio-psychological characteristics and environmental necessities of the elderly are addressed with the concept of aging in place (AIP) in various areas, such as architectural and spatial design, gerontechnology, and UX/UI design [1,2,3,4], which emphasizes support for well-being from a multidimensional perspective. AIP involves aging with familiar people in a familiar area [5], which is related to the desire for settlement and emotional attachment to the individual’s residence [3,6]. AIP practices allow the elderly to maintain connectivity with existing social networks and known support services; however, the role of residential space in AIP is important. The ability of the elderly to stay in their environment intersects with the quality of the residential space [7], and the elderly spend more time at home because they want to heal and recover in a personal space [8]. Accordingly, the environmental and architectural sectors are seeking effective and efficient support for AIP, and the possibility of building a sustainable and innovative residential environment has been discussed while minimizing the help of experts and institutions owing to community networks and smart homes [9].

The smart home service (SHS) for the elderly manages air quality and energy efficiency to promote the convenience of the residential environment, welfare, and safety [10], which supports the convenience of household activities and daily activities. In addition, receiving the benefits of medical and care facilities at home, such as recognizing, predicting, and responding to the risk situation of the elderly, and remotely sharing data with the medical institution in charge is essential. In addition, healing and recovery at home and satisfying experiences are important for the elderly’s desire for settlement. Although the SHS ensures independent living and economic efficiency in housing, it shows a slight lack of consideration for emotional care, such as depression and loneliness [11,12,13].

This study focuses on positive experiences with nature at home, suggesting that SHSs should move toward supporting this type of approach. Previous studies [14,15,16,17], which investigated human biological reactions related to natural experiences or exposure to nature, emphasized that the environment and elements of nature are sources that help strengthen human immune and physiological functions. This comes from the human instinctively nature-loving mind, biophilia, which has evolved from nature [18]. Biophilic design, on which the biophilia hypothesis is based, is the practice of integrating nature into architecture and design to improve health and well-being [19]. Until now, discussions on biophilic design have focused on the planning of a real, touch-based physical environment [20,21,22]; however, it requires links with technology and service sectors to strengthen modern urbanization and contact with nature in an aging society. Previous studies [23,24,25,26] have shown that the absence of natural elements in the physical environment leads to negative psychological states, such as fatigue and depression, and is likely to expose humans to diseases such as high blood pressure, diabetes, and heart disease; therefore, a specific discussion on the support service of biophilic experiences of smart home technology for the elderly and how to apply it are required.

Therefore, the purpose of this study was to compare and analyze the preferences of the elderly and other age groups through a survey on the support service of the biophilic experience in the smart home. Further, the appropriate type and composition of biophilic experience-based SHS are suggested for the elderly at home.

The remainder of this paper is organized as follows. First, through prior research and literature, we examine the residential environment for the elderly, smart homes, and the elderly’s perception of SHS and technology and discuss the characteristics and necessity of biophilic design experience. Second, we reorganized the SHS of the three biophilic experiences and related sensors and devices, focusing on our previous research [27,28]. In this study, we developed a survey instrument for biophilic experience-based SHS. Third, a survey instrument was used to analyze the needs of the elderly differentiated from the non-aged, and the preferences of adults over the age of 20 were surveyed through an online survey. Fourth, we derived the type of biophilic experience-based SHS by conducting a factor analysis of the collected data. In addition, after classifying the respondents’ ages into the elderly and non-aged, independent two-sample *t*-tests were conducted to analyze the items that showed a significant difference in the average preference among groups. Finally, based on the results of the survey analysis, we derived the biophilic experience-based SHS for the elderly and proposed it along with applicable sensors and devices.

This study goes beyond the physical environmental planning of previous biophilic design studies and discusses the linkage with smart home technology. In addition, this study provides a theoretical basis for expanding the concept of biophilic design by seeking SHS plans for the elderly based on positive experiences with nature.In particular, this study reorganized the biophilic experience-based SHS by focusing on the needs of the elderly and improved previous studies by deriving statistically significant service types through a preference survey.Analyzing the biophilic experience needs of the elderly, who are different from other age groups, and suggesting service configurations considering their preferences greatly contributes to the improvement of SHS quality and satisfaction of the elderly.The results of this study provide evidence for services that should be considered first in the early stages of service development and contributes to the application and commercialization of houses by suggesting the sensors and devices required for service development.

## 2. Literature Review and Related Works

### 2.1. Elderly Residential Environment and Smart Home

In old age, there is a possibility of physical illness, cognitive decline, loss and weakening of social roles, and economic difficulties [29]. When recognizing aging, emotional complications, such as emotional anxiety and depression, tend to increase [30,31]. Aging-related changes are sensitive not only to physical health and social environments but also to the physical environment, specifically to the negative impact of living conditions, and the elderly are more vulnerable than young adults [32]. The environmental docility hypothesis means that the lower the individual’s ability, the more influence the environment has on suggesting the person’s function [33], and thus, it puts an emphasis on planning the physical environment according to the individual’s specific requirements. This is an essential approach in the residential environment because the elderly spend more time at home as their mobility and activity decrease, and thus they are relatively more likely to be exposed to stress factors because of the weakening of their health functions [34]. In addition, studies [8,35] related to the experience of healing and recovery of the elderly have suggested that many retired elderly prefer the home environment to recover and heal. This should be noted, along with previous studies [36], which propose that the meaning and specific requirements of the living environment for the elderly are different from those of other age groups. When planning a residential environment, the differentiated necessities of the elderly who are different from other age groups must be satisfied.

The purpose of AIP is to satisfy the demands and abilities of the elderly who want to live independently in suitable housing [37]. AIP is especially important for the elderly in terms of the meaning of the residential environment and the attachment to the place. As the importance of AIP has emerged, active research has been conducted for establishing a residential environment to respond to changes in old age. As a solution that has been widely discussed in recent years, smart homes are a personal support for the elderly and have developed as a connecting medium [38]. Smart homes are residential spaces that integrate communication networks, sensors, home appliances, and smart devices [39] and provide services that meet the necessities of residents. The ultimate goal of a smart home is to improve the well-being and QoL of residents [40]; for elderly residents, it is crucial to provide an SHS that focuses on prevention and promotion rather than solutions to aging problems [41].

This study reviewed the core contents of previous research related to smart home planning and composition for the elderly and the technology utilization plan and analyzed the possibility of supporting the elderly physically, mentally, socially, and economically. Table 1 summarizes the review and analysis of previous studies related to smart homes for the elderly.

Smart homes are undergoing technical convergence to build automated systems that can interact with occupants, devices, and the environment. Related studies have discussed the development of integrated services using advanced technologies, such as smart applications (apps), robotics, artificial intelligence (AI), natural language understanding (NLU), and immersive technology, focusing on the Internet of Things (IoT).

Technical elements include various types of sensors, devices, and monitoring systems, along with a central control hub. However, despite various technological convergences, the scope of SHS provision for the elderly focuses on supporting energy efficiency and physical functions, such as indoor environment control and automation [44,53,54,55]. However, despite various technological convergences, the scope of SHS provision for the elderly focuses on supporting energy efficiency and physical functions, such as indoor environment control, automation [46,47,48,49], and remote response in case of emergency [42,43]. On the other hand, in the case of SHS related to mental health, studies dealing with practical support services at the level of detecting or predicting dementia and depressive states are insufficient [52,58,59]. Therefore, this study emphasized the need for psychological satisfaction and positive emotional management of the elderly in their house and sought an SHS plan that considers the requirements of the elderly from a bio-friendly perspective.

### 2.2. Elderly Perception on Smart Home Service and Technology

To support the improvement of elderly QoL via an SHS, it is important to understand the perceptions and effects of the elderly on the technical aspects at the planning stage. Through a prior study [40,60], this study examined the awareness of the elderly towards SHSs and technology in terms of satisfaction, pleasure, usability, necessity, reliability, and economic feasibility.

The satisfaction and enjoyment of the elderly are related to mental stimulation and multisensory activities, using which they hope to learn, explore, or interact [41]. The need for these stimuli reflects the process of overcoming the isolation caused by the decline in cognitive and sensory functions [61]. A prior SHS study [62] on the leisure and satisfaction of the elderly showed they were satisfied with experiencing the natural environment through virtual reality (VR) or augmented reality (AR) devices, and they expressed an overall preference for the experience of immersion. They preferred the Nintendo Wii Sport Game, which was easy to learn and use; after game training, their bond with their grandchildren and their physical, social, and psychological well-being substantially improved [63]. A study [52] that examined the satisfaction of the elderly with social robots shaped like real dogs found no difference between experiences with real dogs or robots; all participants showed improved function of expression with high satisfaction. The elderlies’ perception of the usability of smart home technology is related to a reasonable level of familiarity and comfort [64], which is also related to how they interact with the sensors and devices and their manipulation [65]. The elderly wanted understandable and repetitive rules and an intuitive sense of use, such as familiar graphics or wearable devices [66], and they recognized the usefulness of tablet PCs and apps [61]. This stems from the familiarity with smartphones and preference for large screens, suggesting the possibility of acceptance of smart apps, general user interfaces (GUIs), and multimodal interfaces. However, they expressed resistance to inconveniences interfering with routine tasks, such as periodic replacement and charging of wearable sensors [67,68]. Another concern was if they use personal alarms or portable emergency buttons, they may lose or forget to use them [69]. In terms of necessity, the majority of the elderly understand the benefits of an SHS and technology and expect it to be beneficial [70]. However, they tend to regard it as necessary for older people other than themselves [71]. More importantly, they reject an excessive level of health care or unnecessary technologies, rather than health care or technologies themselves. Ultimately, they deny these health care or technologies when they are thought of as patients [64]. In particular, after using SHS, the elderly were found to be conscious of other people’s views regarding the use of specialized equipment [69], and the fears associated with it can have an emotionally negative impact. Many of the elderlies’ negative perceptions of reliability have emerged as a matter of personal privacy and security in their homes [68,70,72,73,74]. Elderly people are aware of the ongoing management benefits of monitoring services; however, they have expressed anxiety that privacy can be abandoned or violated [73], and they have psychological resistance to camera-based monitoring, continuous filming, and location tracking [68,74]. In addition, they may distrust automated systems, which consider the recognition of emergency situations, the possibility of errors in emergency contacts, etc., and show low reliability in the clarity of the scope of recognition and detection [70]. Finally, even though elderly people in the community consider using a new SHS, cost remains a primary concern [75]. Building a smart home requires the integration of various sensors and devices with new software technologies, which can lead to high initial costs. After retirement, the elderly are sensitive to economic feasibility due to their limited income; therefore, the low price versus performance cost of the service, or the high initial cost of sensors and devices, are the most decisive factors in their rejection of SHSs.

Therefore, based on this analysis, a service plan was established. Further, the selection of appropriate technologies and an integrated composition of sensors and devices are required.

### 2.3. Biophilic Experience and Smart Home Service

Psychologist Fromm [76] mentioned “biophilia” as the concept of human love for life and all living things, which was later popularized as the biophilia hypothesis by biologist Wilson [18]. The biophilia hypothesis is based on the biological necessities of mankind that evolved from nature; owing to mankind’s “adaption” to nature over a long period of time, the biological function of man is most optimized when it cooperates with nature [19]. Access and exposure to the natural environment can greatly contribute to human health and well-being through numerous mediated pathways and direct effects [17,77,78]; in particular, it contributes substantially to the QoL of the elderly in terms of physical, mental, and social anxiety [16,79,80]. However, biophilia also has a negative impact on humans in an environment that lacks nature as such an environment can likely cause diseases in humans, such as fatigue, depression, hypertension, diabetes, heart disease, and even cancer [23,24,25,26]. Therefore, the elderly, whose ability to physiologically adapt to environmental stressors is reduced, are more vulnerable to aging in urban environments where nature is relatively scarce [81]. The importance of biophilic design based on the biophilia hypothesis is emerging to manage the necessity of nature and the dense urban environment. Biophilic design studies the effects of natural environmental properties, such as light, air, color, materials, and plants, on human physiology and psychology and is a spatial design strategy to improve personal experiences that occur when interacting with these characteristics [20]. Until now, discussions on biophilic design have focused on physical environment planning based on reality [27]; to reinforce experiences with nature in urban residential environments and an aging society, connection with technology and service fields is required. This study emphasizes that smart home technologies and services should support experiences with nature for positive emotions and the satisfaction of residents.

Kellert [82] presented three important experiences in the practice of biophilic design and categorized them into direct experience of nature, indirect experience of nature, and spatio-temporal experience. Direct experience of nature refers to actual contact with the natural environment and elements; indirect experience of nature is an analogue of nature or experience based on its features and processes. Finally, the experience of space and place focuses on spatial solutions to experience the spatial characteristics of a positive natural environment, as well as simulated natural environments, such as VR and AR. This is because the effects of virtual nature have been proven in studies on human biological responses to videos or images of nature, or in VR and AR [62,83,84]. Recently, attempts have been made to support various experiences with nature by linking biophilic design and smart technology; however, it is important to analyze users’ requirements for specific SHS development and efficient application. Therefore, we reorganized the biophilic experience-based SHS based on the previously discussed contents and the results of our previous studies [27,28] to analyze the biophilic SHS preference of the elderly. Table 2 lists the SHS, related devices, and sensors for the three biophilic experiences.

Sensors and devices for biophilic experience-based SHSs are classified as outdoor monitoring sensors, indoor monitoring sensors, vital sensors, multimodal sensors, controllers, smart devices and appliances, and immersive displays [27]. Outdoor/indoor monitoring collects weather information, indoor air quality, and light environment data and provides an opportunity for occupants to be exposed to the natural environment by being connected to controllers and actuators, smart devices, and appliances. A vital/multimodal sensor is used to recognize the state and situation of the resident and helps with the resident’s emotional recovery and intuitive interaction with virtual natural elements. An immersive display is a hologram projector that includes a head-mounted display (HMD)/eye glass display (EGD), and a screen-type display. It is a visual device that can be immersed in VR/AR elements and virtual natural environments. Recently, immersive displays and multisensory actuators, which support the five senses, such as hearing, tactile, and olfactory, have been developed. Therefore, a service plan differentiated from the existing SHS must be discussed.

## 3. Materials and Methods

### 3.1. Data Collection

This study conducted an online survey to compare and analyze the level of access to an SHS for biophilic experience for the elderly and the sense of preference among other age groups. The subjects of the survey were adults aged 20 years or older living in Korea, with the same ratio of allocation of the samples to the entry group of the elderly (aged 50–70 years or older) and other age groups (ages 20–49). Before the survey on preferences began, demographic data (age, gender, type of housing, etc.) were collected, along with basic information about SHSs. This was used as a screening question to minimize bias in the responses of participants without prior knowledge of smart homes or their services. The question, “Do you know what smart home technologies and services are?” was asked with five possible responses: “I don’t know”, “I know a little”, “I know in general”, “I know well”, and “I am already using smart home technology and services”. Respondents who selected the option “I don’t know” were filtered and were not able to participate in the subsequent survey, and all other respondents participated in the survey after becoming familiar with the description of the biophilic design experience. Finally, 250 sets of survey data were collected. The general characteristics of the respondents were the same, as presented in Table 3.

There were 102 men (40.8%) and 148 women (59.2%), with 63 aged 50–59 years (25.2%). The number of residents of apartments was the highest at 183 (73.2%), as the types of apartments or multi-family homes became common owing to Korea’s cultural characteristics and the limitations of density and population scale.

### 3.2. Construct Operationalization

In this study, overlapping or integratable content based on Table 2 was reorganized and used as a tool for a survey of preferences for the biophilic experience of SHSs, with a total of 28 questions; all questions were rated on a 5-point Likert scale (1 = “not preferred”–5 = “very preferred”). Details of the questionnaire used in the study are provided in Appendix A Table A1. The questions in the survey reflected repeated tests and improvements for clarity to verify its reliability and internal consistency, as shown in Table 4. Reliability refers to the degree to which the same result is obtained when repeatedly measured, and the internal consistency method is generally used the most [97,98]. The internal consistency method measures the degree of consistency between several items for measuring the same concept. In the SPSS program, the internal consistency method is calculated using Cronbach’s alpha coefficient. Cronbach’s alpha coefficient is distributed between 0 and 1 based on the correlation coefficient between items, and the larger the average correlation coefficient between items, the higher the alpha value [99,100]. In general, Cronbach’s alpha values above 0.60 are considered relatively reliable, and values above 0.70 are considered highly reliable [101]. The value for the survey was 0.840, thereby verifying the internal reliability of the questionnaire.

### 3.3. Data Analysis

In this study, 250 data points collected from the online survey were set as a valid sample group of n, and IBM SPSS Statistics 25.0 ver. was used for statistical analysis. The data was analyzed as follows: First, the measurement of Cronbach’s alpha confirmed the internal consistency of the questionnaire and analyzed the results of the descriptive analysis for the overall response data. Second, to categorize the biophilic experience based SHS, a factor analysis was performed on the values of the variables in the 28 items. Principal component analysis and varimax rotation were used for factor analysis, and the names of the factors and types of services were derived based on the contents of the service extracted by each factor. Third, independent samples t-test was employed to analyze the difference in the average preference between the elderly and non-elderly, including the entry layer of the elderly. Finally, this study comprehensively discusses how to plan and organize the biophilic experience based SHS considering the stated preferences of the elderly.

## 4. Results

### 4.1. Types of Biophilic Experience-Based SHS

Based on the results of the survey, a factor analysis was conducted for categorizing SHS based on biophilic experiences. The Kaiser–Meyer–Olkin (KMO) value for the extraction of the factor is 0.892, which indicates a good correlation between variables. In addition, Bartlett’s test of sphericity showed a significance probability of 0.00, which proved the suitability of the factor analysis model. In this study, the eigenvalue in the early stages of the factor analysis was 1.0% or higher, and for factor loading, six factors were extracted based on a value of ≥0.4 [102]. The explanatory power for all factors was 56.91%. Table 5 lists the types of services derived through factor analysis and the results of the descriptive statistics.

In this study, based on the results of the factor analysis, the types of services for six factors were named ‘Immersion and interaction with nature’ (A), ‘Management of well-being and IEQ’ (B), ‘Changes in weather and natural light’ (C), ‘Communion with animals and plants’ (D), ‘Simulation of natural features’ (E), and ‘Natural process and systems’ (F). The types of immersion and interactions with nature (A) include technical elements that directly or indirectly interact with the resident or physical environment or immerse in a virtual environment to provide services. Services of immersion and interaction with nature include immersive content, 3D virtual objects, and sound similar to reality, providing residents with sensory stimulation and satisfaction. The types of well-being and IEQ management (B) include customized services for the residents in terms of lighting, air quality, temperature, humidity, etc., in their home, which include the inflow of external natural elements to the interior to create a pleasant environment. In addition, this allows the control of the indoor environment according to the residents’ biological and psychological conditions, and it provides training content that considers the level of physical and mental functions. The types of changes in weather and natural light (C) provide a multi-sensory experience of changes in nature over time, including services such as the collection of rainwater, sound of rain, view of outside nature, and inflow and production of natural light. The type of interaction between animals and plants (D) maintains the appropriate habitat environment of the creatures in the home, provides relevant information, and supports emotional communication through real animal-like social robots or 3D objects. The simulation of natural features (E) is a type of service that produces unique and locational characteristics of nature, such as sunlight, shadows, or waves, reflected between the branches. In addition, it provides services for variable control over interior elements that symbolize nature, such as colors and patterns. Finally, the type of natural processes and systems (F) provides beneficial services to both residents and ecosystems in residential areas. The natural process refers to the interaction between plants, animals, and the environment [103]. Smart home technology helps activate interactions such as photosynthesis, pollination, and decomposition, which results in services for energy generation and waste disposal.

### 4.2. Preference for a Biophilic Experience-Based SHS

Because of the descriptive statistics on SHS based on the preference of biophilic experience, the average of all items was 3.97. Table 6 lists the average value for these 28 services. The results of the analysis of average preference revealed that the services with the highest preference were the controllable digital wall (4.28) and type A, followed by indoor air quality monitoring and automatic ventilation (4.21), microbial applications for bioenergy production and disposal systems (4.19), new renewable energy production and rainwater recycling (4.17), and the provision of physical and cognitive training based on virtual nature (4.09).

To identify the requirements of the elderly that differed from other age groups, respondents were divided into the elderly (aged ≥ 50; n = 125) and non-elderly (aged 20–49; n = 125) groups. Table 6 lists the results of the *t*-test for both the groups.

The t-test of SHSs based on the preference of the biophilic experience showed that the total average of the elderly (4.02) was higher than that of the non-elderly (3.92). There was a statistically significant difference between the 17 service items. Among them, the non-elderly showed a higher average in 6 services (v6, v9, v15, v17, v22, and v24), while the elderly showed a higher average in 11 services (v2, v4, v7, v12, v14, v16, v23, v25, v26, v27, and v28). Elderly people preferred intuitive interaction services, such as a virtual sky on the ceiling, natural elements that respond to touch and gesture, and social robots for communication. In particular, this showed a high stated preference in terms of the experiences of immersion in nature, such as listening to the sound of rain or exposure to natural light. In addition, new services that are not easily experienced at home were found to be more pursued than daily convenience services, such as automatic management of animals and plants or notification of weather information. Services for physical, biological, and emotional well-being (v4, v23, v25) and natural processes (v27, v28) exhibited a relatively high average of preference and a high level of significance (*p* < 0.000). Additionally, these services exhibited a high awareness of their healing and environmental benefits in the home.

### 4.3. Composition of Biophilic Experience-Based SHSs for the Elderly

In this study, statistically significant items were extracted through t-test analysis, which prioritized items when planning biophilic experience-based services for smart homes for the elderly, as shown in Table 7. The item was first extracted based on the abovementioned average value (≥4.02) of elderly preferences among the 11 items that showed higher preference compared to the non-elderly responses. Second, even if the results of the t-test were insignificant, they exceeded the overall average value (≥3.97) and included items that met the average values (≥3.92, ≥4.02) or more of the two groups. This was used to investigate the general use of biophilic experience-based SHS. Finally, one item (v7) was excluded from the first condition, and a total of 15 services were proposed, including five items (v3, v5, v10, v20, v21) that meet the second condition.

This study also proposed the composition of sensors and devices of biophilic experience-based SHSs to use at home. This composition was based on the support for 15 services based on the contents examined in a previous study, and the details are presented in Table 8.

Outdoor sensors and controllers were included in the A, B, and C services; indoor sensors were used for all types of services; and vital sensors were used for B services. Multimodal sensors were used with immersive displays to provide A, B, and E services, and smart devices and appliances were used in all types of services, except E.

Outdoor monitoring sensors collect various environmental data, such as the passage of weather and sunlight and the sound of rain. This includes temperature, humidity, wind volume, wind direction, and heliostat. A heliostat is a device that includes a mirror, usually a plane mirror, which turns so as to keep reflecting sunlight toward a predetermined target, compensating for the sun’s apparent motions in the sky [104]. Indoor monitoring sensors measure air quality, illumination, and energy consumption. In particular, lighting-based measurement technologies, such as LiDAR sensors, are appropriate for collecting necessary data in terms of occupant privacy and security, through monitoring a room without a camera or scanning the 3D space. Environmental sensors play an important role in identifying the elements of immersion in the home or controlling the environment, which contributes to supporting the health and economic characteristics of the elderly by improving the opportunities for contact with nature. Vital and multimodal sensors are used to analyze occupant data, enabling precise measurements of ultra-small and wearable sensors. In addition, it supports the intuitive interaction of the smart environment with the occupants, such as monitoring various diseases and symptoms with minimal vital signs and building responsive interfaces that are easy to access and control. The controller of the smart home provides experience of direct and indirect natural factors, such as natural lighting and air quality, as well as natural color and sound, based on automated systems, to immediately respond to environmental stress factors. Smart devices and appliances combine several technologies, which are used to maximize immersion in nature by connecting with displays such as IVGC and haptic actuators. Further, as the method of interaction becomes more diverse, it is used to train physical and cognitive functions. In addition, smart technology from a bio-friendly perspective mimics the functions of nature or promotes natural processes, contributing to improving the quality of the environment and preserving the indigenous ecosystem. An immersive display is necessary to produce life-like nature, and its visual exposure to nature offers more advantages over other senses. In particular, it is important to consider the development of display-based content when planning the service of biophilic experience, noting that there is a high awareness of elderly preference for large display and visual interaction devices.

## 5. Discussion and Conclusions

Providing SHSs and technology to satisfy the experience of the elderly in a residential environment is an important part of the improvement of personal QoL and environmental welfare, which provides access to public health macroscopically by supporting everyday places that promote health and well-being. Thus, this study proposed the composition of a biophilic experience-based SHS considering the requirements of the elderly and the entry layer of the elderly group. Twenty-eight services were categorized through a survey of preferences for a biophilic experience-based SHS, and differences in preferences between elderly and non-elderly individuals were compared and analyzed. Finally, the composition of sensors and devices was proposed and expanded, along with 15 services of five types.

First, we categorized biophilic experience based SHS types, and the following order was determined, from highest to lowest: ‘Natural process and systems’ > ‘Simulation of natural features’ > ‘Management of well-being and IEQ’ > ‘Changes in weather and natural light’ > ‘Immersion and interaction with nature’ > ‘Communion with animals and plants’. Considering the results of the overall preference, when applying the biophilic experience-based SHS, the digital wall to use elements of symbolic and metaphorical representations of nature, such as colors and patterns, should be considered, and it is important to provide them so that they can be easily controlled according to the preferences of the occupants. In addition, an automated system is required to allow natural wind and air flow indoors according to wind volume and wind direction, and this system should be adjusted according to the level of indoor air quality. The open window that makes outside nature visible requires a blocking function if necessary; natural light must be provided through artificial sunlight in spaces where it is difficult to secure illumination depending on the structure of the home.

Second, because of the analysis of the entry layer of the elderly group and their preference for a biophilic experience-based SHS, there was a high preference for natural immersion based on intuitive interaction and immersive display. This is similar to the results of a prior study [59,64] related to simple usability and the elderly’s preference for display. In particular, they seek new services differentiated from the existing SHS, such as experiencing nature through NUL interfaces, including touch and gesture, rather than daily natural experiences, such as automatic plant management or weather information notifications. In addition, even without direct interaction, they were found to have a high preference for visual immersion, such as a skylight display to show the virtual sky and wall display to produce waves and waterfalls.

Third, the elderly showed a high preference for enhancing customized functions based on the natural environment and the provision of healing elements of nature based on biological and emotional conditions. Further, they pursued a recovery environment based on multi-sensory experiences, such as temperature and humidity control, natural sound, and color. Considering the negative perceptions [60,66] of the elderly regarding the excessive level of medical services, they are generally considered to have a positive perception of healing and recovery services through biophilic experience. In other words, this shows that services should support sensors and devices that allow the elderly to experience multi-sensory elements of nature and that the creation of a healing environment, in terms of prevention and the development of biophilic contents for the elderly, are necessary.

Finally, the elderlies’ preferred type of service are natural processes and systems. Further, they pursue economically efficient maintenance and management of housing, such as via solar power, recycling rainwater, and creating natural process-based bioenergy. The elderly feel economically burdened by the use of SHSs, and the cost of initial construction can have a significant impact on their decision to reject the services. The bio-friendly home system provides clear benefits compared to the initial costs for the elderly, such as solving the problems of waste disposal and energy consumption and contributing to the preservation of the indigenous ecosystems. Furthermore, device cost and space efficiency should be considered through the integration and collaboration of the composition of the sensors and devices proposed in this study.

This study proposed a new biophilic experience in the field of SHS for the elderly and addressed the technical perception reflecting the demands of the elderly. Moreover, in contradiction to existing SHS that focused on responding to disease, this study focused on preventing disease from a biophilia perspective. The results of this study provide references for market conversion and commercialization of bio-friendly ideas in the development of SHSs and healthcare services for the elderly. In particular, this study contributes to the utilization of practical applications in priority services and promotes the composition of sensors and devices by type to facilitate the satisfactory experience of the elderly at home. However, this study addresses the composition of services, focusing on hardware, and the geographical distribution of those surveyed was limited. In addition, because this study focused on the positive experiences of the elderly with nature in their daily living space, future research is necessary to organize services according to the physical, mental, and social characteristics of the elderly by securing a sufficient scope of sampling and number of samples. Furthermore, it is important to expand the results of this study by actively determining the framework and realization of biophilic experience-based SHS.

## Figures and Tables

**Table 1 sensors-21-05108-t001:** Review of previous studies related to smart homes for the elderly.

Ref.	Purpose	Technical Tools	Support			
			Phys.	Psych.	Soc.	Eco.
[42]	Elder fall detection system using smart tiles	Sensing floor; force sensor; three-axis accelerometer	●			
[43]	Fall detection and related information extraction through LiDAR ^1^ sensor-based monitoring system	LiDAR sensor; machine learning; EMG (accelerometer, electromyogram) sensor	●			
[44]	Suggestion of smart home services for health status, activity, and environmental perception of the elderly	Health monitoring system; ambient assisted living system; SMAF (Functional autonomy measurement system)	●			●
[45]	Development of T-shirt for physiological condition measurement, such as heart rate, breathing, pulse, and body temperature	Internet of things (IoT); electrocardiogram (ECG) sensor; wearable sensor	●			
[46]	Real-time gait monitoring system for stroke prediction service	IoT based gait monitoring sensors; vital sign monitoring; machine learning based disease prediction model; smart phone application	●			
[47]	Real-time health monitoring system for early stroke prognostics and the management of post-stroke treatment	Eye-mask embedded portable EEG(electroencephalography); medical ontology-based health advisor service	●			
[48]	Proposed smart home services platform for diabetes monitoring	Environmental sensor; wearable sensor; application; IoT	●			●
[49]	Monitoring chronic diseases in older adults and suggesting a framework for predicting depression and diabetes (analysis of irregular activity patterns)	Binary sensor (eating, bowel movements, sleep, activity, conversation, night activities, etc.); smart appliance; web-based healthcare applications	●	●		
[50]	Development and usability evaluation of digital frame ‘Tlatoque’ that supports family communication and customized SNS activities for the elderly	Ambient social network system; virtual reality (VR); touch screen	●		●	
[51]	Evaluation of Social Support for Older Adults using Social Robot ‘Paro’	Robotics technology; natural language understanding (NLU); artificial intelligence (AI)		●	●	
[52]	Comparison and analysis of elderly depression symptoms with dog social robot ‘Aibo’ and real pets	Robotics technology; AI; touch sensor		●	●	
[53]	Presentation of green remodeling and smart home framework for health and energy efficiency of the elderly	IoT; environmental sensor; automatic controller and actuator	●			●
[54]	Automatic energy consumption control system according to user lifestyle	Deep learning; IoT	●			●
[55]	Smart Home Control Service Plan based on audio technology (sweet-home project)	Audio-based interaction technology; multimodal sound	●		●	●
[56]	Suggestion of smart assistive architecture for elderly and disabled care environment control	Robotic system; IoT; multimodal interface; augmented reality (AR)	●			●
[57]	Physical, mental, and social well-being platform and application based on lifestyle	Multimodal sensor; big data; activity recognition; social media technology; wearable sensor-based AR	●	●	●	●
[58]	Development of depression detection framework through analysis of indoor location-based elderly activity patterns	Passive infra-red (PIR) sensor; modal-neural network; Bayesian network		●		
[59]	Development of virtual environment contents related to past memories and memories for the elderly	VR; AR; image-based rendering		●		

^1^ LiDAR, or light detection and ranging, is a popular remote sensing method used for measuring the exact distance of an object on the earth’s surface.

**Table 2 sensors-21-05108-t002:** Biophilic experience-based service configuration of smart home for the elderly.

Biophilic Experience-Based SHS			Sensors	Device	Literature Source
Direct experience of nature	S.1	Adjustment of louvers and curtains according to the amount of sunlight and its direction	Outdoor monitoring	Louver/curtain controller	[27,85,86]
	S.2	Real-time tracking of sunlight paths and inflow of reflected light	Outdoor monitoring	Reflect panel controller	[28,82]
	S.3	Periodic natural ventilation and window opening	Indoor monitoring	Window controller	[27,53]
	S.4	Induction of natural ventilation and airflows according to the wind volume and direction	Outdoor monitoring	Window controller	[27,87]
	S.5	Automatic temperature and humidity control according to the weather	Outdoor monitoring	HVAC ^1^ controller	[27,53]
	S.6	Measurement of indoor air quality level and automatic ventilation	Outdoor/indoor monitoring	Window/HVAC controller	[27,88]
	S.7	Green-wall automation management	-	Automation green-wall	[27,28,89]
	S.8	Provision of social robots in the shape of puppies	Multimodal sensor	AI social robot	[27,51,52]
	S.9	Automatic management of water quality and temperature, food, etc., in water tanks	-	Smart aquarium	[27,28]
	S.10	Real-time plant status check and automatic management	-	Smart plants growers	[27,28]
	S.11	Provision of windows that are remotely transparent	-	Smart window display	[27]
	S.12	Provision of real-time information on outside weather and conditions	Outdoor monitoring	Smart window display	[27,41]
	S.13	Rainwater collection and sound provision	Outdoor monitoring	Stereo controller	[27,90]
Indirect experience of nature	S.14	Immersive display showing the occupant’s desired natural landscape	-	Smart window display	[27,41,91]
	S.15	Lighting display showing a virtual sky	-	Smart window display	[27,41,91]
	S.16	Simulation of seasonal environments according to climate and of natural sounds	Outdoor/indoor monitoring	Wall display, Stereo controller	[27,77]
	S.17	Provision of virtual water objects and sounds such as waterfalls and waves	Indoor monitoring	Immersive display, Stereo controller	[27]
	S.18	Provision of 3D virtual animal objects	Indoor monitoring	Immersive display	[27,41]
	S.19	Lighting control according to simulation of virtual objects and background	Indoor monitoring	Lighting controller	[27]
	S.20	Predicting the emotional state of the elderly and providing healing elements	Vital/multimodal sensor	Immersive display, stereo/lighting controller	[92]
	S.21	Provision of natural sound by occupant context	Vital/multimodal sensor	Stereo controller	[27,28]
	S.22	Provision of schedule and weather information according to occupant’s lifestyle	Outdoor monitoring	Smart window display	[85]
	S.23	Controllable virtual wall patterns and textures based on nature	Multimodal sensor	Wall display	[27]
	S.24	Provision of artificial sunlight by illumination	Indoor monitoring	Lighting controller	[27,28]
	S.25	Provision of various virtual light shapes and shadows	Indoor monitoring	Projector lamp	[27]
	S.26	Monitoring of occupant’s physiological condition and energy efficiency	Vital/indoor monitoring	HVAC controller	[53,85]
	S.27	Interior environment control by occupant’s condition	Vital monitoring	Lighting/stereo controller	[27]
Experience of space and place	S.28	Provision of VR/AR contents for nature experience	Multimodal sensor	Immersive display	[27,93]
	S.29	Provision of management APP and prediction for occupant’s physical and mental health level	Vital/multimodal sensor	Smart phone	[63,94]
	S.30	Provision of nature-based contents for physical and mental training	Multimodal sensor	Immersive display	[27,63,94,95]
	S.31	Virtual natural elements that respond to touches and gestures	Multimodal sensor	Immersive display, stereo controller	[27,91]
	S.32	Waste collection and natural disposal	Indoor monitoring	Smart kitchen/toilet	[27,96]
	S.33	Natural energy reproduction and rainwater recycling	Outdoor monitoring	NERS ^2^/PVT ^3^/RRS ^4^	[27,28]
	S.34	Provision of microbial bioenergy	Indoor monitoring	Biogas generating system	[28]

^1^ Heating, ventilating, and air conditioning, ^2^ Natural Energy Reproduction System, ^3^ Photovoltaic Thermal, ^4^ Rainwater Recycling System.

**Table 3 sensors-21-05108-t003:** Descriptive statistics of demographic information.

Characteristics		Number (N)	N %
Age	20–29	30	12.0
	30–39	49	19.6
	40–49	46	18.4
	50–59	63	25.2
	60–69	40	16.0
	70 and above	22	8.8
Gender	Male	102	40.8
	Female	148	59.2
Household type	Apartment	183	73.2
	Townhouse	42	16.8
	Detached house	8	3.2
	Office hotel	6	2.4
	The others	11	4.4
Total		250	100.0

**Table 4 sensors-21-05108-t004:** Internal consistency of the used questionnaire.

Item	Number (N)	Cronbach’s Alpha
Preference	28	0.840

**Table 5 sensors-21-05108-t005:** Results of descriptive statistics and factor analysis.

Type	Factors: M (SD)	Variable		M	SD	Factor Loading	Eigen Values	Dispersion Ratio (%)
A	Immersion and interaction with Nature: 3.93 (0.09)	v13	Display window showing the desired natural scenery	3.85	0.83	0.734	3.540	13.616
		v19	Provision of soundscapes according to simulation of virtual nature	3.85	0.77	0.673		
		v26	Natural immersive elements that respond to touch and gestures	4.00	0.76	0.599		
		v15	Experience of immersed seasonal environment based on VR/AR	4.00	0.76	0.585		
		v14	Skylight display showing a virtual sky according to the weather	4.01	0.90	0.576		
		v18	Lighting control for immersion environment in nature	3.83	0.82	0.530		
		v24	VR/AR contents for experience of virtual natural environment	4.01	0.68	0.496		
B	Management of well-being and IEQ: 4.04 (0.11)	v06	Automatic management system of green-wall based on air purification plants	3.88	0.81	0.703	3.424	13.171
		v04	Automatic temperature/humidity control according to physiological conditions and weather	4.02	0.84	0.669		
		v03	Induction of natural ventilation and air flow according to the air volume and direction	4.08	0.68	0.664		
		v05	Indoor air quality monitoring and automatic ventilation	4.21	0.76	0.646		
		v23	Provision of natural elements (sound, lighting, etc.) according to the occupant’s emotional state	3.97	0.95	0.511		
		v25	Provision of physical and cognitive training based on virtual nature	4.09	0.84	0.509		
C	Changes in weather and natural light: 3.96 (0.07)	v12	Provision of rain sounds from the outdoors	3.94	0.86	0.654	2.057	7.910
		v10	Transparent switchable windows for viewing	3.98	0.84	0.603		
		v02	Real-time tracking of sunlight paths and inflow of reflected light	3.97	0.81	0.576		
		v21	Provision of artificial sunlight according to illuminance	4.08	0.71	0.575		
		v11	Provision of real-time weather and outdoor status information	3.87	0.72	0.529		
		v01	Control of louvers and curtains according to the amount of sunlight and its direction	3.93	0.69	0.499		
D	Communion with animals and plants: 3.76 (0.15)	v07	AI social care robot similar to an animal	3.81	0.90	0.757	2.050	7.884
		v09	Real-time plant automatic cultivation and condition management	3.95	0.72	0.669		
		v17	Provision of augmented 3D animal objects (butterfly, bird, dolphin, etc.)	3.65	0.92	0.518		
		v08	Automatic management of fish tank (water quality and temperature, feeding, etc.)	3.63	0.84	0.487		
E	Simulation of natural features: 4.06 (0.20)	v20	Controllable digital wall (pattern/color/etc.)	4.28	0.75	0.685	1.965	7.559
		v22	Lighting projector for natural light shapes and shadows	3.95	0.82	0.592		
		v16	Simulation of waterfalls and waves using walls or floors	3.94	0.87	0.556		
F	Natural Process and Systems: 4.18 (0.01)	v28	New renewable energy production and rainwater recycling	4.17	0.69	0.524	1.760	6.768
		v27	Microbial applications for bioenergy production and disposal system	4.19	0.69	0.514		
	Total			3.97	0.15		-	

**Table 6 sensors-21-05108-t006:** T-test on the preference mean of the elderly and non-elderly.

Type	Factors	Variable	Non-Elderly		Elderly		*F*	*t*
			M	SD	M	SD		
A	Interaction and immersion in nature	v13	3.78	0.89	3.91	0.76	5.367	−1.225
		v19	3.83	0.79	3.86	0.76	1.025	−0.327
		v26	3.83	0.80	4.17	0.68	3.567	−3.575 ***
		v15	4.17	0.68	3.83	0.80	3.567	3.575 ***
		v14	3.78	0.92	4.23	0.82	1.444	−4.052 ***
		v18	3.82	0.81	3.85	0.83	0.120	−0.308
		v24	4.18	0.45	3.83	0.82	41.739	4.212 ***
B	Management of IEQ and well-being	v06	4.04	0.73	3.73	0.86	7.419	3.095 **
		v04	3.66	0.79	4.38	0.73	0.940	−7.484 ***
		v03	4.13	0.72	4.02	0.64	5.371	1.208
		v05	4.26	0.75	4.17	0.77	0.099	0.915
		v23	3.75	0.97	4.19	0.87	1.975	−4.475 ***
		v25	3.77	0.83	4.42	0.72	2.185	−6.573 ***
C	Changes in weather and natural light	v12	3.77	0.82	4.11	0.85	0.108	−4.016 ***
		v10	3.93	0.87	4.02	0.81	0.933	−0.903
		v02	3.85	0.81	4.09	0.78	0.660	−2.376 *
		v21	4.01	0.72	4.15	0.71	1.068	−1.151
		v11	3.95	0.77	3.79	0.66	0.310	1.759
		v01	3.97	0.71	3.90	0.67	0.472	0.827
D	Communion with animals and plants	v07	3.66	0.81	3.96	0.90	3.811	−5.096 ***
		v09	4.06	0.64	3.84	0.78	4.559	2.401 *
		v17	3.86	0.87	3.43	0.92	0.109	3.809 ***
		v08	3.67	0.88	3.58	0.79	3.369	0.831
E	Simulation of natural features	v20	4.21	0.82	4.36	0.68	1.887	−1.603
		v22	4.02	0.78	3.88	0.82	0.635	3.400 **
		v16	3.79	0.87	4.08	0.84	0.009	−3.024 **
F	Natural process and systems	v28	4.01	0.80	4.34	0.52	3.741	−3.844 ***
		v27	4.05	0.82	4.33	0.49	14.988	−3.276 **
	Total		3.92	0.17	4.02	0.25	-	-

* *p* < 0.05, ** *p* < 0.01, *** *p* < 0.000.

**Table 7 sensors-21-05108-t007:** Biophilic experience based SHS considering the preferred results.

Type	Biophilic Experience Based SHS		Total		Non-Elderly		Elderly	
			M	SD	M	SD	M	SD
A	v26	Natural immersive elements that respond to touch and gestures	4.00	0.76	3.83	0.80	4.17	0.68
	v14	Skylight display showing a virtual sky according to the weather	4.01	0.90	3.78	0.92	4.23	0.82
B	v04	Automatic temperature/humidity control according to physiological conditions and weather	4.02	0.84	3.66	0.79	4.38	0.73
	v03	Induction of natural ventilation and air flow according to the air volume and direction	4.08	0.68	4.13	0.72	4.02	0.64
	v05	Indoor air quality monitoring and automatic ventilation	4.21	0.76	4.26	0.75	4.17	0.77
	v23	Provision of natural elements (sound, lighting, etc.) according to the occupant’s emotional state	3.97	0.95	3.75	0.97	4.19	0.87
	v25	Provision of physical and cognitive training based on virtual nature	4.09	0.84	3.77	0.83	4.42	0.72
C	v12	Provision of rain sounds from the outdoors	3.94	0.86	3.77	0.82	4.11	0.85
	v10	Transparent switchable windows for viewing	3.98	0.84	3.93	0.87	4.02	0.81
	v02	Real-time tracking of sunlight paths and inflow of reflected light	3.97	0.81	3.85	0.81	4.09	0.78
	v21	Provision of artificial sunlight according to illuminance	4.08	0.71	4.01	0.72	4.15	0.71
E	v20	Controllable digital wall (pattern/color/etc.)	4.28	0.75	4.21	0.82	4.36	0.68
	v16	Simulation of waterfalls and waves using walls or floors	3.94	0.87	3.79	0.87	4.08	0.84
F	v28	New renewable energy production and rainwater recycling	4.17	0.69	4.01	0.80	4.34	0.52
	v27	Microbial applications for bioenergy production and disposal system	4.19	0.69	4.05	0.82	4.33	0.49

**Table 8 sensors-21-05108-t008:** Composition of biophilic experience based SHSs for the elderly.

Services		Sensor				Device		
		Outdoor Monitoring	Indoor Monitoring	Vital	Multimodal	Controller	Smart Devices and Appliances	Immersive Display
A	v26		LiDAR		Touch/motion	Stereo cont.	Haptic actuator	Wall display, hologram project, HMD/EGD
	v14	Temperature/humidity						Skylight display
B	v04	Temperature/humidity	Temperature/humidity	TPR ^1^		HVAC cont.		
	v03	Wind volume/direction	CO2/VOCs/PM-10, etc.			Window cont.		
	v05		CO2/VOCs/PM-10, etc.			HVAC cont.		
	v23		LiDAR	TPR/ECG		Stereo/lighting cont.		
	v25			TPR	Touch/motion/voice		IVGC ^2^/haptic actuator	HMD/EGD
C	v12	Audio/microphone				Stereo cont.		
	v10						Smart glass	
	v02	Heliostat sensor				Reflect panel cont.		
	v21		Luminance			Lighting cont.	Artificial sunlight lamp	
E	v20		LiDAR		Touch/motion/voice			Wall display
	v16							Wall display
F	v28		Energy monitoring				PVTS/RRS	
	v27		Energy monitoring				BES ^3^	

^1^ Temperature/Pulse/Respiration, ^2^ Interactive Video Game Console, ^3^ Bioenergy System.

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
