# Peer review of "A Preference-Driven Smart Home Service for the Elderly’s Biophilic Experience"

_sensors, 2021, doi:10.3390/s21155108_

Round 1

Reviewer 1 Report

This study aimed to evaluate the Biophilic features of Smart Home Service through the survey. I have the following suggestions.

  1. Please rewrite the title of this paper.
  2. The abstract needs to be improved by adding significant parametric findings of this study. The result part of the abstract should be improved.
  3. The contributions of this study should be added as bulleted points in the introduction section.
  4. Grammar should be revised, and article readability should be improved.

Example: Line142: Pereption>>Perception

  1. If possible, some tables can be shifted to the supplementary material section.
  2. References can be improved by adding related articles in elderly health monitoring, sleep monitoring, and disease prognostics applicable in the smart home framework.
      1. Elderly health monitoring is a crucial part of a smart home. The author should include related research, https://doi.org/10.1109/ICCE46568.2020.9043098, https://doi.org/10.1007/978-3-030-20476-1_11, https://doi.org/10.1007/978-3-319-96098-2_27
      2. Sleep monitoring and disease prognostics are widely studied for smart home applications. The author should mention related research, such as https://doi.org/10.1109/ACCESS.2020.3040437
  3. A comparative table should be added in the discussion section comparing other studies.

Author Response

-

This study aimed to evaluate the Biophilic features of Smart Home Service through the survey. I have the following suggestions.

We are grateful for the careful review of this paper. We value your opinion and have attempted to reflect it.

Point 1

Please rewrite the title of this paper.

Response 1

The title has been revised according to your comment.

Point 2

The abstract needs to be improved by adding significant parametric findings of this study. The result part of the abstract should be improved.

Response 2

We supplemented and revised the abstract by adding the results of this study to reflect your comment.

Point 3

The contributions of this study should be added as bulleted points in the introduction section.

Response 3

We added the contribution and novelty of this study in the introduction section.

Point 4

Grammar should be revised, and article readability should be improved. Example: Line142: Pereption>>Perception

Response 4

In response to your comment, we have improved the overall readability of the manuscript by employing an academic proofreading service company (Please refer to the attached file of the editing certificate).

Point 5

If possible, some tables can be shifted to the supplementary material section.

Response 5

Table 4 has been changed to Appendix A based on your comments.

Point 6

References can be improved by adding related articles in elderly health monitoring, sleep monitoring, and disease prognostics applicable in the smart home framework.

1. Elderly health monitoring is a crucial part of a smart home. The author should include related research,

https://doi.org/10.1109/ICCE46568.2020.9043098

https://doi.org/10.1007/978-3-030-20476-1_11

https://doi.org/10.1007/978-3-319-96098-2_27

2. Sleep monitoring and disease prognostics are widely studied for smart home applications. The author should mention related research, such as

https://doi.org/10.1109/ACCESS.2020.3040437

Response 6

We reviewed additional previous studies on disease prediction and continuous management of the elderly, and through this, Section 2.1 and Table 1 were supplemented.

Point 7

A comparative table should be added in the discussion section comparing other studies.

Response 7

As recommended, we added references [46] and [47] for comparison with other studies and considered a comparative table. However, this study focuses on analyzing the preference for biophilic experience rather than specific diseases of the elderly. Other studies have dealt with disease management and service realization for the elderly; however, considering the relative insufficiency in this study, the future research direction was included in the conclusion.

Reviewer 2 Report

The paper deals with biophilic indoor environment design in smart homes for the elderly. It is, in general, well organized and explained but some very specific concepts should be better detailed, such as Cronbach's alpha. 

Author Response

Point 1

The paper deals with biophilic indoor environment design in smart homes for the elderly. It is, in general, well organized and explained but some very specific concepts should be better detailed, such as Cronbach's alpha.

Response 1

We value your opinion and are grateful for your positive review of this manuscript. Referring to your comments, we described the concept of internal consistency and Cronbach’s alpha in Section 3.2.